# Prospecting the UNESCO World Heritage Site of Müstair (Switzerland)

Jona Schlegel [1,*], Geert J. Verhoeven [1,2], Patrick Cassitti [3], Alois Hinterleitner [1,4], Klaus Löcker [1,4], Hannes Schiel [1,4], Christoph Walser [5], Thomas Reitmaier [5] and Wolfgang Neubauer [1,6]

1   Ludwig Boltzmann Institute for Archaeological Prospection and Virtual Archaeology, Hohe Warte 38, 1190 Vienna, Austria; geert.verhoeven@archpro.lbg.ac.at (G.J.V.); alois.hinterleitner@archpro.lbg.ac.at (A.H.); klaus.loecker@archpro.lbg.ac.at (K.L.); hannes.schiel@archpro.lbg.ac.at (H.S.); wolfgang.neubauer@archpro.lbg.ac.at (W.N.)
2   Faculty of Arts and Philosophy, Department of Archaeology, Ghent University, Sint-Pietersnieuwstraat 35, 9000 Ghent, Belgium
3   Stiftung Pro Kloster St. Johann, via Maistra 18, 7537 Müstair, Switzerland; patrick.cassitti@muestair.ch
4   ZAMG—Zentralanstalt für Meteorologie und Geodynamik, Hohe Warte 38, 1190 Vienna, Austria
5   Archaeological Service of the Canton of Grisons, Loëstrasse 26, 7001 Chur, Switzerland; christoph.walser@adg.gr.ch (C.W.); thomas.reitmaier@adg.gr.ch (T.R.)
6   Vienna Institute for Archaeological Science, University of Vienna, Franz-Klein-Gasse 1, 1190 Vienna, Austria
*   Correspondence: jona.schlegel@archpro.lbg.ac.at

**Abstract:** The Benedictine Convent of Saint John at Müstair is a UNESCO World Heritage Site located in the eastern part of Switzerland close to South Tyrol's border (Italy). Known as a well-preserved Carolingian building complex housing Carolingian and Romanesque frescoes, the convent has received much academic attention. However, all research activities so far have been concentrated on the area enclosed by the convent's walls, even though the neighbouring fields to the east and south are also part of the convent's property. This paper reports on the archaeological magnetic and ground-penetrating radar surveys of these areas, executed as part of a pilot project exploring the convent's immediate environment. At present, these fields are used for agriculture and located on a massive alluvial fan of the mountain stream Valgarola. Dense geophysical sampling revealed an intricate network of distributary channels with stream and mudflow deposits, constituting a natural border of the convent's territory. In addition to different field systems, a newly discovered broad pathway appears to be an original Roman road. Numerous structural elements, mapped within the convent's walls, could be attributed to specific building phases. Over 40 large and deep burial shafts, arranged in three rows, were discovered outside the convent's burial ground. Their specific design and arrangement are characteristic of early medieval burials, such as those of the 6th century Lombards on the edge of the eastern Alps.

**Keywords:** archaeological prospection; convent; burials; ground-penetrating radar; magnetometry; Middle Ages; Longobards; Müstair; Roman road; Switzerland; water management

## 1. Introduction

The Benedictine Convent of Saint John at Müstair (Canton of Grisons, Switzerland) was built around AD 775 during the reign of Charlemagne and is presently located near the border of South Tyrol (Italy). The convent has been the subject of archaeological, historical, and architectural research, which started in 1884 with Joseph Zemp and Robert Durrer. These scholars examined and documented the convent buildings, which led to discovering its Carolingian and Romanesque wall paintings. These well-preserved frescoes were the main argument to date the original building complex to the Carolingian period. Additionally, they secured a UNESCO World Heritage Site label in 1983 [1].

Significant restoration and renovation activities started in 1964, which lead to the creation of the "Pro Kloster St. Johann in Müstair" foundation. Thanks to the foundation's

long-term commitment to restore and preserve this outstanding monument, systematic building recordings and architectural studies saw daylight [1]. Furthermore, an excavated Bronze Age settlement, Neolithic and Iron Age finds, and a Roman posthole building fed the hypothesis of Müstair's continuous settlement history [2].

The convent's intramural area equals 1.7 hectares, including the Carolingian church and the Holy Cross Chapel, the residential Plantatower, the residence of a former bishop, and two rectangular courtyards. To the west, cloisters, two entrance towers, and agricultural buildings surround the courtyard. A medieval/recent graveyard is attached to the Carolingian church in the east. The rest of the convent's property consists of arable land and meadows (Figure 1).

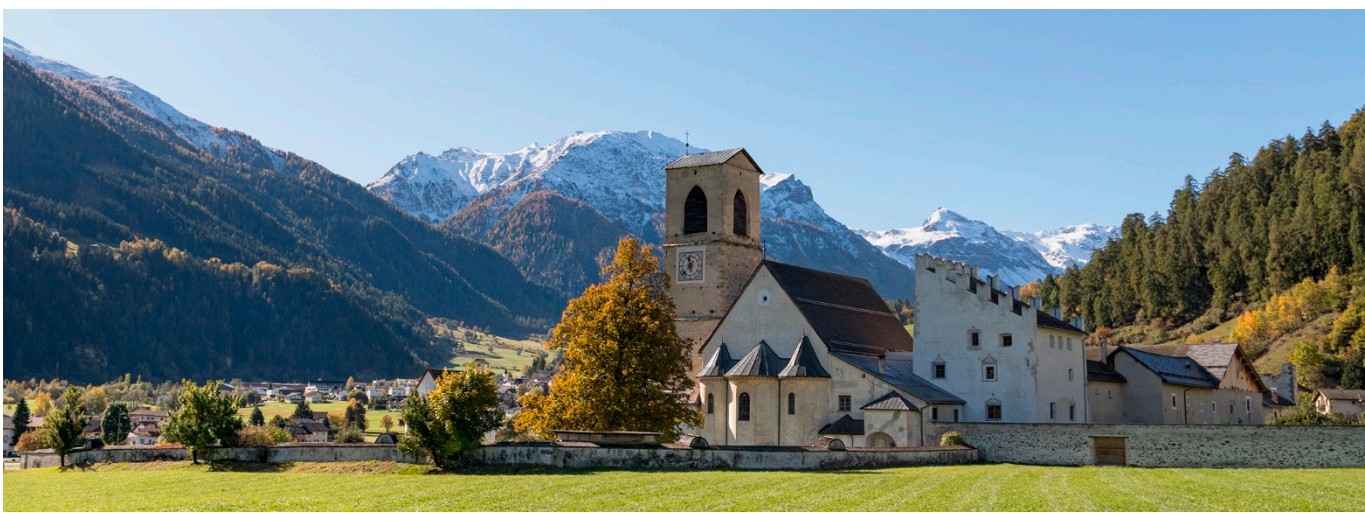

**Figure 1.** Panoramic picture of the Convent of Saint John in Müstair from the east to the west (photograph by K. Löcker, adapted by G. Verhoeven).

The convent complex in the valley of Müstair is situated above the floodplain of the Rom stream and bordered on the northwest by the Ruinatscha mountain slope (Figure 2 at the foot of the Piz Terza mountain, also known as Urtirolaspitz. The Valgarola stream in the east comes from the Avigna valley and splits upon its exit of the valley into three different streams which offer freshwater to the convent and the villages of Müstair (Switzerland) and Taufers (Italy). The Valgarola flows from the mountains onto the Rom plain and has over time created a relatively gently sloping (circa 6°) fan consisting of typical stream, flood, and mudflow deposits. This big alluvial fan separates Müstair and Taufers, thereby forming the border between Switzerland and Italy (Figure 2) [3,4].

Jürg Leckebusch conducted the first geophysical research at the convent. He acquired ground-penetrating radar (GPR) profiles on the outside wall of the Plantatower [5] and an unpublished area east of the convent. This paper reports on the geophysical data acquired during a two-day campaign in October 2019 and showcases the first results of the interpretive mapping. This pilot study was initiated by the Ludwig Boltzmann Institute for Archaeological Prospection and Virtual Archaeology (LBI ArchPro) in cooperation with the foundation "Pro Kloster St. Johann in Müstair" and the Archaeological Service of the Canton of Grisons, aiming to evaluate the potential of a mechanically noninvasive prospection of the convent's immediate environment (which also falls within the UNESCO perimeter).

Thus far, three areas have been covered (Figure 3): (1) a meadow area east of the convent walls (almost 2.3 ha motorised GPR and 2.8 ha magnetometry), (2) the graveyard inside the convent's walls (0.4 ha handheld GPR), and (3) a section of the tarred road south of the convent plus an adjacent meadow (totalling circa 0.5 ha motorised GPR). The next sections detail the acquisition, processing, and interpretative mapping of these data.

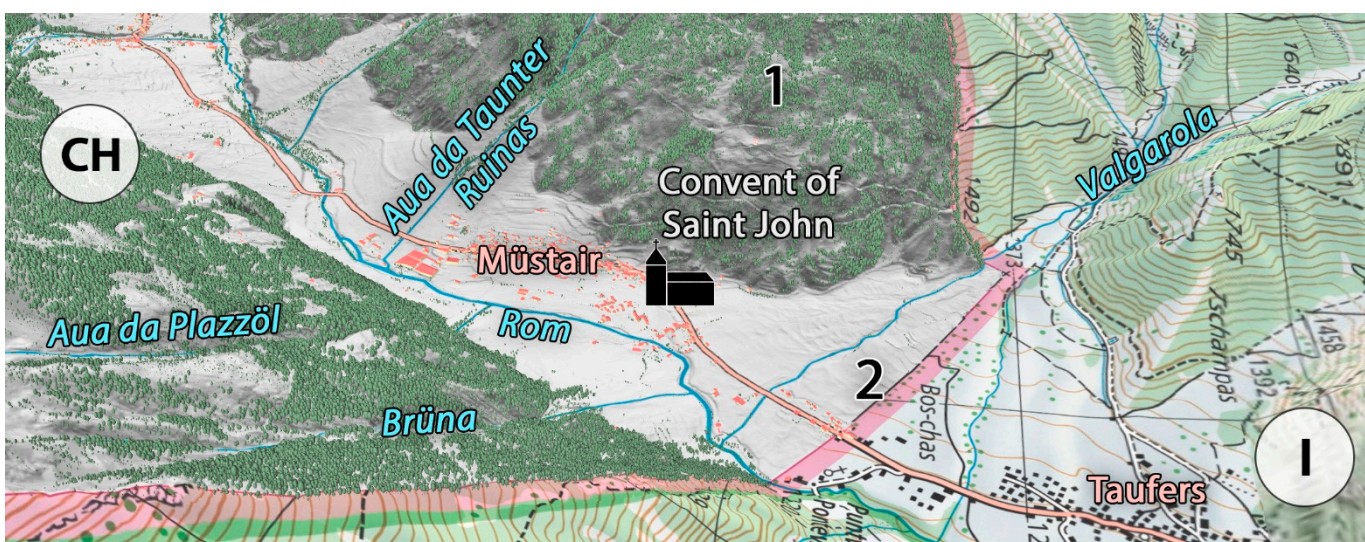

**Figure 2.** The topographical situation around the Convent of Saint John at Müstair. The figure depicts the main river Rom and its tributaries: Aua da Taunter Ruinas, ("Aua da" meaning "water from"), Aua da Plazzöl, and Valgarola. In addition, there is also a small creek from the Brüna valley. (**1**) indicates the Ruinatscha slope, while (**2**) indicates the big alluvial fan.

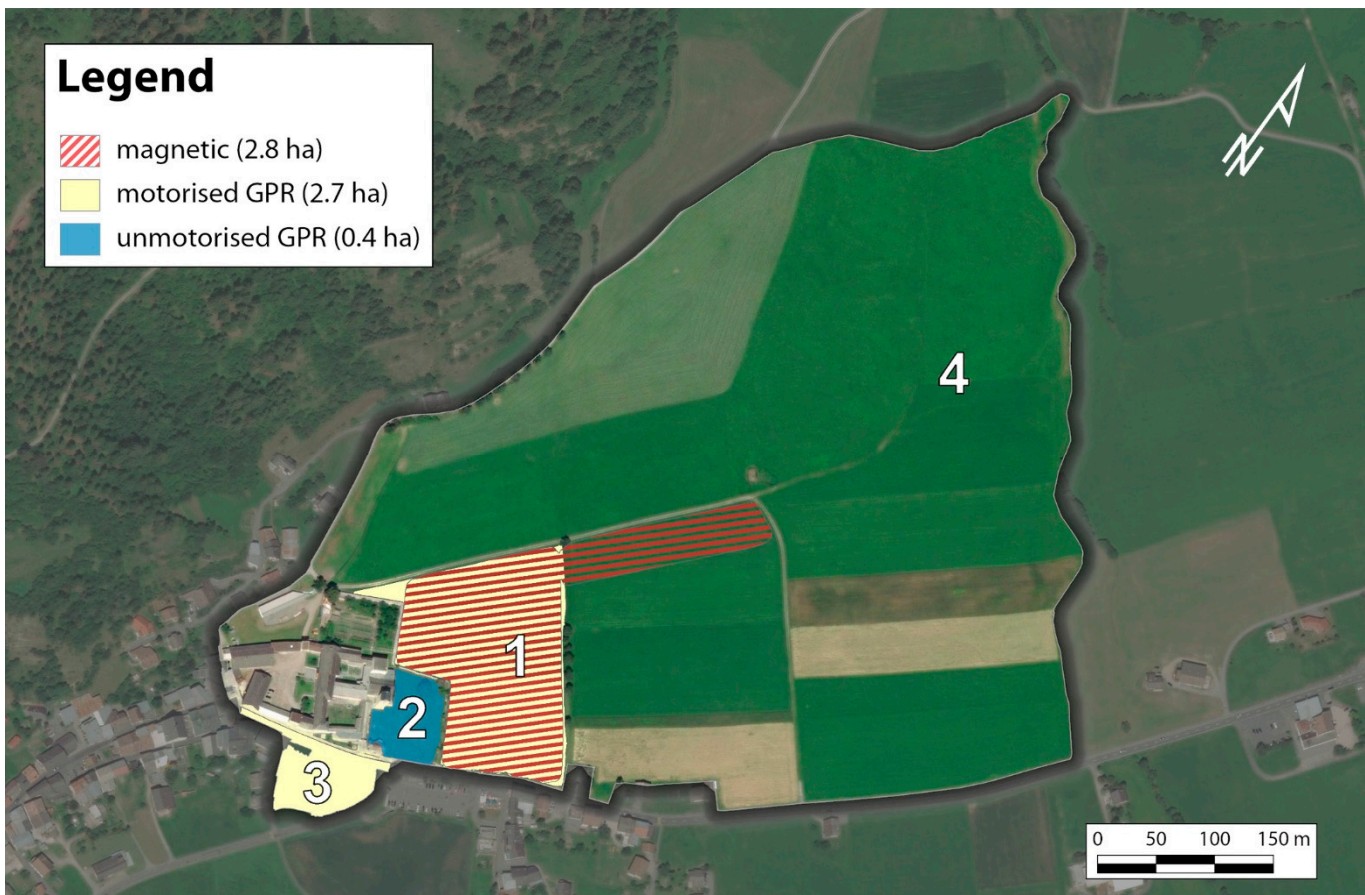

**Figure 3.** The prospected areas (**1** to **3**) and remaining convent fields to prospect in the future (**4**). Background map source: Esri, Maxar, GeoEye, Earthstar Geographics, CNES/Airbus D.S., USDA, USGS, AeroGRID, IGN and GIS User Community.

## 2. Geophysical Surveys

The bulk of the GPR data were acquired with a 16-channel 400 MHz MALÅImaging Radar Array (MIRA) GPR system developed by MALÅGeoscience (now Guideline Geo). The MIRA antenna box is front mounted on the hydraulic fork of a Kubota BX2350 tractor, which features a JAVAD-based, real-time kinematic–global navigation satellite system (RTK-GNSS) solution delivering a planimetric positional accuracy of circa 3 cm (Figure 4C). The real-time navigation software LoggerVis developed by the LBI ArchPro facilitates the system's positioning, while the raw radar data are recorded and stored using Guideline Geo's software (MIRASoft). This combination provides fast, large-area, and useful data acquisition in the field [6], allowing for on-the-fly identification of potential data flaws and holes. In total, this MIRA system covered some 2.7 ha (Figure 3, areas 1 and 3) with an 8 cm crossline spacing. As these measurements are time triggered, the survey speed is chosen so that the inline trace spacing never exceeds 4 cm.

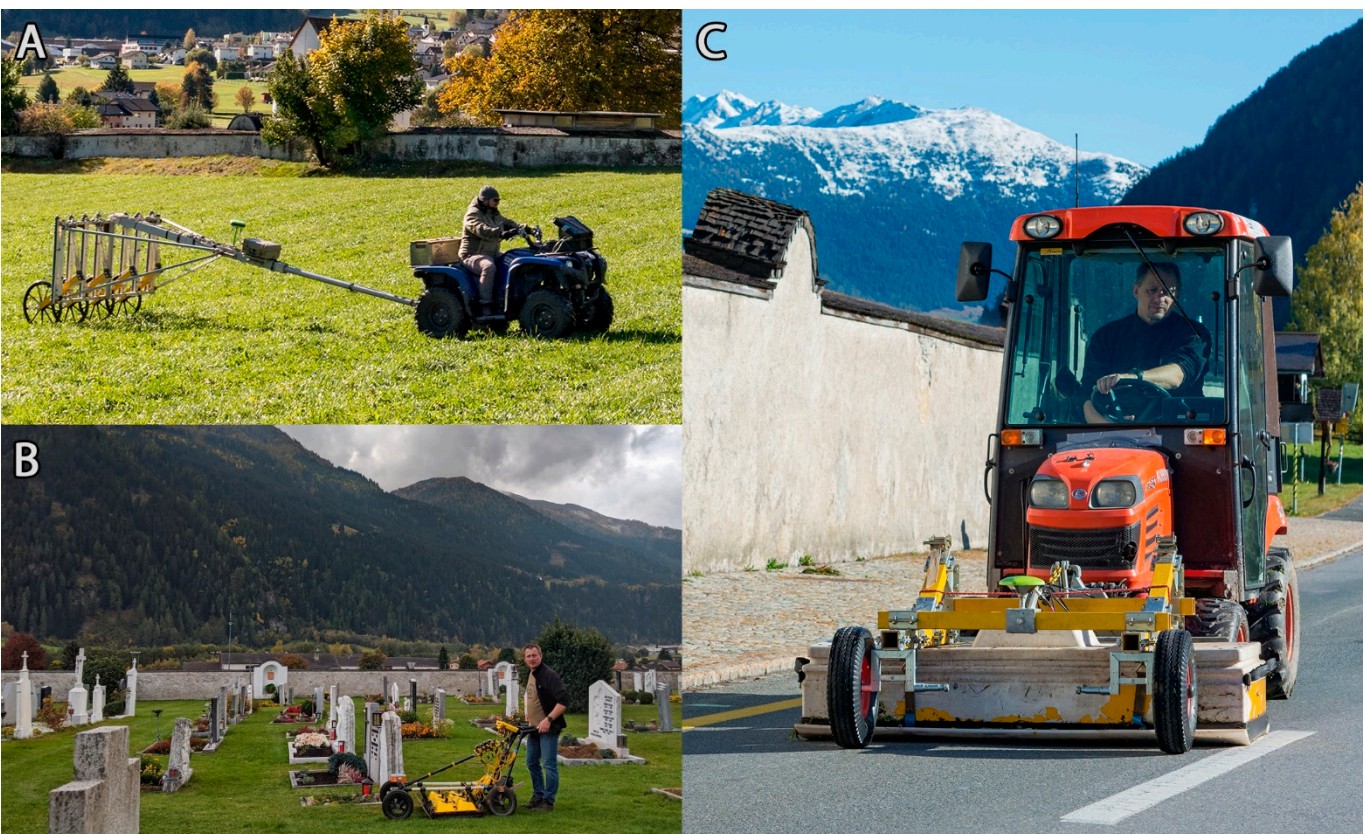

**Figure 4.** The three geophysical prospection systems applied in Müstair: (**A**) motorised multisensory fluxgate magnetometer, (**B**) handheld three-antenna GPR, and (**C**) motorised 16-channel GPR. Photograph (**A**) by Klaus Löcker, (**B**) by Jona Schlegel, and (**C**) by Geert J. Verhoeven.

A total of 0.4 ha of the current graveyard (Figure 3, area 2) was covered with a manual three-channel GPR system based on three 500 MHz Sensors & Software pulseEKKO®PRO antennae mounted in an odometer-enabled SmartCart (Figure 4B), enabling a 25 cm crossline and 5 cm inline sampling. Accurate positioning of the data was mainly established using an RTK-based data stream visualised in LoggerVis. Some data were collected along a grid of parallel lines marked on the ground due to the RTK signal shielding by a tree. Data recording was taken care of by the Sensors & Software data logging unit. Both GPR surveys took place when the soil was relatively dry and thus well suited for prospection.

The meadow to the west of the convent (Figure 3, area 1) was surveyed for 2.8 ha with a quad bike towing a nonmagnetic cart equipped with eight Foerster FEREX CON650 gradiometer probes (Figure 4A) and a Panasonic Toughbook running LoggerVis for real-



time position visualisation and data acquisition. The system allows for a 25 cm crossline spatial sampling, while the time-based triggering yields a maximum inline sampling distance of 10 cm. Positioning of the magnetometer data is established through a JAVAD TRIUMPH-1 GNSS receiver with about 3 cm RTK planimetric positional accuracy at a 5 Hz sampling rate.

## 3. Data Processing

The processing of the densely sampled magnetic and GPR data was carried out in ApSoft (consider [6] for a detailed account of all processing steps), bespoke software developed through a collaborative effort by the LBI ArchPro and the Central Institute for Meteorology and Geodynamics (ZAMG). The raw GPR data processing results in vertical, two-dimensional reflection profiles and a resampled three-dimensional data block with a planimetric cell size of 8 cm by 8 cm and 5 cm in depth. This data cube was subsequently cut into horizontal depth slices of varying thickness (5 cm, 10 cm, 20 cm, 50 cm). Every one of these slices encodes the georeferenced magnitudes of the reflected radio waves. Their visualisation is accomplished through greyscale images in which increasing brightness represents locations with decreasing reflection magnitudes. Since the composition and moisture content of the soil influence the propagating speed of the radar signal, speed analysis of the local soil matrix is crucial to approximate realistic depth values of the obtained signal reflections. Even though the Müstair data have been velocity corrected, deviations up to 20% in the depths of the GPR slices remain possible because the radio waves' propagation velocity depends on many parameters that can be very variable spatially (whereas the correction assumes a globally uniform propagation speed).

The magnetometer data were displayed as greyscale images with $-8$ nT and $+12$ nT as cutoff values to enable the interpretative mapping of relevant anomalies. All processed prospection data were combined with georeferenced aerial photographs, historical maps, and other geographical data into a GIS. Their analysis and the creation of interpretative maps was carried out in ESRI's ArcMap 10.3 and ArcGIS Pro 2.5, together with the LBI ArchPro's ArchaeoAnalyst extension for ArcMap. The next section presents these results.

## 4. Results and Discussion

As Dodge et al. [7] state, mapping entails visualising, classifying, representing, and communicating information about places. In archaeology, these maps usually represent anthropogenic patterns distilled from various data and information sources, such as field diaries, prospection and excavation results, historical documents, imagery, and historical maps. All these sources need their specific interpretative approaches, from which the interpretative mapper then combines the results into one final document [8]. In doing so, the interpretative map becomes an archaeological narrative that makes the archaeological features, the landscape, and their historical background and formation processes tangible.

The primary goal of the mapping efforts presented here is to sketch a comprehensive picture of the geological situation and place modern, ancient, and prehistoric anthropogenic structures in their landscape context. To that end, the collected survey data and additional information were inserted into a GIS and interpreted in 2.5 and 3 spatial dimensions (i.e., 2.5D and 3D, respectively). All the features detected in the magnetic and GPR data are displayed on the map as polylines or polygons using specific symbology to represent well-defined classes. Interpreting and dating these structures were often possible thanks to the integration with (and georeferencing of) historical sources, such as maps and excavation reports. Modern structures such as drainage pipes were omitted in all maps.

### 4.1. Roads

In the centre of area 1, a 150 m long, east–west oriented solid structure can be identified (Figure 5A). The varying width from 3.6 m to 5.5 m and the length indicates a road.

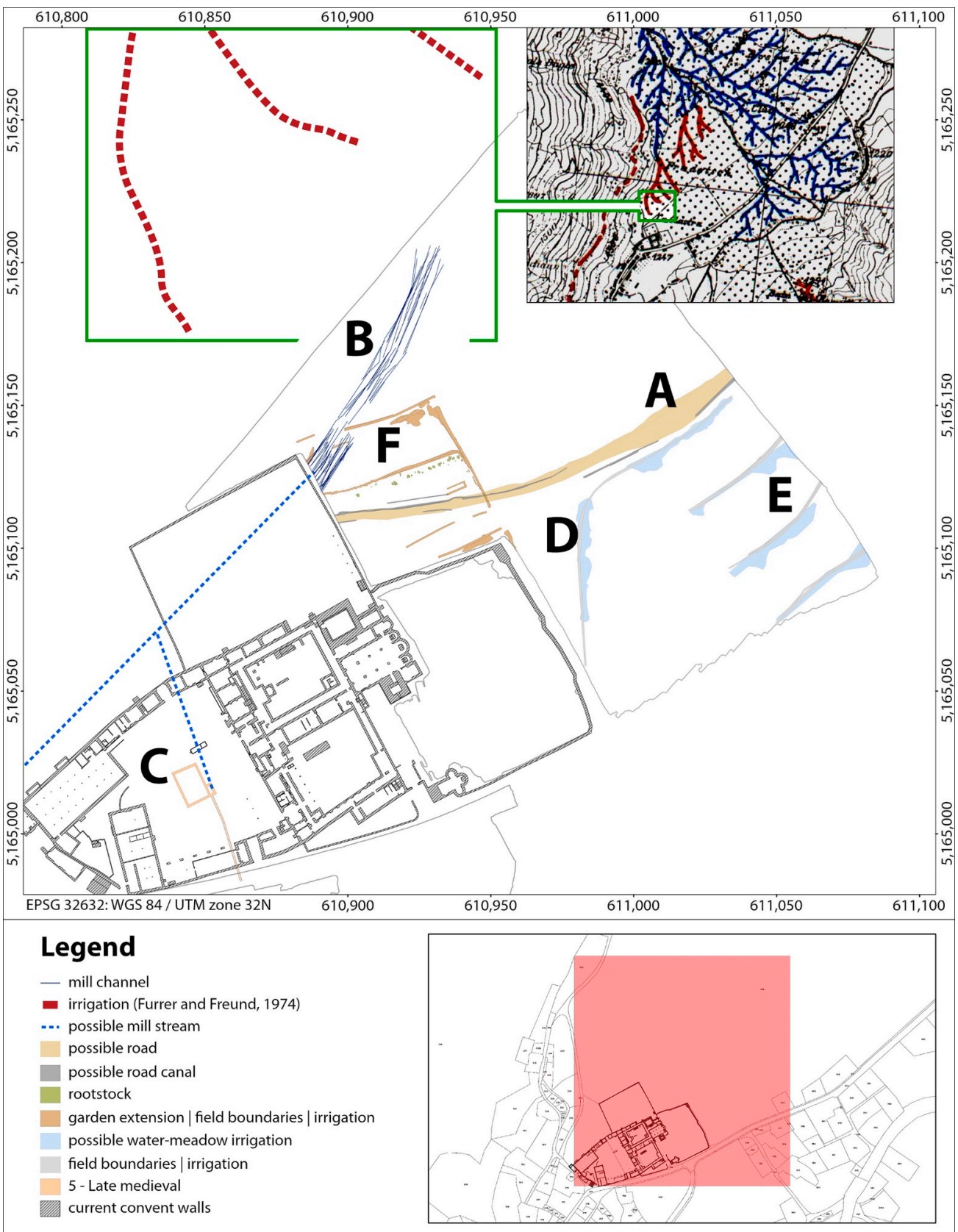

**Figure 5.** Overview of the known historical structures and the new prospection results in area 1 (see the text for the capitalised characters, **A–F**). The upper right insert shows a part of the map mentioned in Furrer and Freund [9].

A smaller linear feature seems to border the road in some places, potentially a roadside ditch or drain. It shows as a highly reflecting structure in the GPR data, indicating a backfill.

The dating of this road is unclear and can only be revealed through excavation. Nevertheless, historians assumed that Müstair acted as a Roman road station before developing into a religious centre under Charlemagne in circa AD 775 [10]. Roman coins [11] and the excavation of a Roman posthole building inside the west court of the convent make the presence of a small Roman settlement such as a hamlet [12] plausible. Even without solid evidence of a Roman road station at the site of the present convent [12], its proximity to the *Via Claudia Augusta* and the recent discovery of Roman settlements in Taufers/Puntweil, Laatsch, and Mals [13] show that this hamlet was at least located within an existing settlement network, which must naturally also have included roads (for a general overview, consider [14–16]).

These observations support the hypothesis that the elongated structure found in the geophysical data might be a Roman road, even more so when considering its characteristic features [17]: drainage ditches that flank a largely linear structure with a width of about 12 Roman feet to almost 20 Roman feet. Given its width, this section could be part of a possible long-distance road across the Ofen Pass west of Müstair [18], branched from the nearby *Via Claudia Augusta* between Augsburg and Verona. At the same time, that road could maybe even function as a *decumanus* of the local Roman land delimitation and division system [19]. At a certain point, this road must have branched into (or crossed) a long-distance road across the Umbrail Pass and the Valtellina Valley south of Müstair. This road to the south could have continued to Como and Milan, the capital of the *Raetia secunda* province in Late Antiquity. In the Early Medieval Period, the capital of the Lombard Kingdom, Pavia, would also have been accessible by this route [12,20,21].

It cannot be excluded that the attested east–west road is of medieval or postmedieval origin. In medieval times, it would have passed the so-called Eginotower (see Section 4.5.3), an excavated structure thought to have served as the Bishop's residence from the 12th to the 14th century [22]. However, the size of the elongated feature and the fact that it points toward the Roman posthole building (and thus the supposed Roman settlement) make this hypothesis much less likely.

### 4.2. Natural Fluvial Features

In survey area 1, the GPR data reveal large meandering reflective structures with different sizes and shapes. They start at a depth of 50 cm beneath the present surface and can be traced further downward in the data. These reflecting structures, which widen as they deepen, could be fluviatile deposits connected to the alluvial fan of the Valgarola mountain stream, which flows into the Rom river near the convent (Figure 2). This hypothesis corresponds to the results of geological borehole drilling, which revealed that the Valgarola deposits are located in the first 14.5 meters below the current walking surface [23].

Fan-shaped debris cones form at locations where a stream emerges from a mountain front. The accumulated sediments—which usually include streamflow and mudflow deposits—often force the stream to split into a system of distributary channels [24], which the GPR data reveals (Figure 6). One broad active channel of almost 60 m contains the braided stream system, which comprises a network of smaller channels separated by islands of gravel and sand deposits. The less reflective structures—visible in the bank areas—may indicate accumulations of coarser sediments.

Fluvial deposits can also be detected in the radar depth slices of survey area 3. It is uncertain how these deposits to the south of the convent relate to those in zone 1, as no fluvial features were detected in survey area 2. Suppose these reflective structures are part of the same braided river stream system detected in survey zone 1. In that case, they must also have largely fallen dry before the Roman period because the hypothesised Roman road (Figure 5A) superimposes all fluviatile deposits. However, at least a part of this alluvial

fan stream system must have remained active for much longer, as the GPR data revealed various channels to rid excess water north of the convent (see next section).

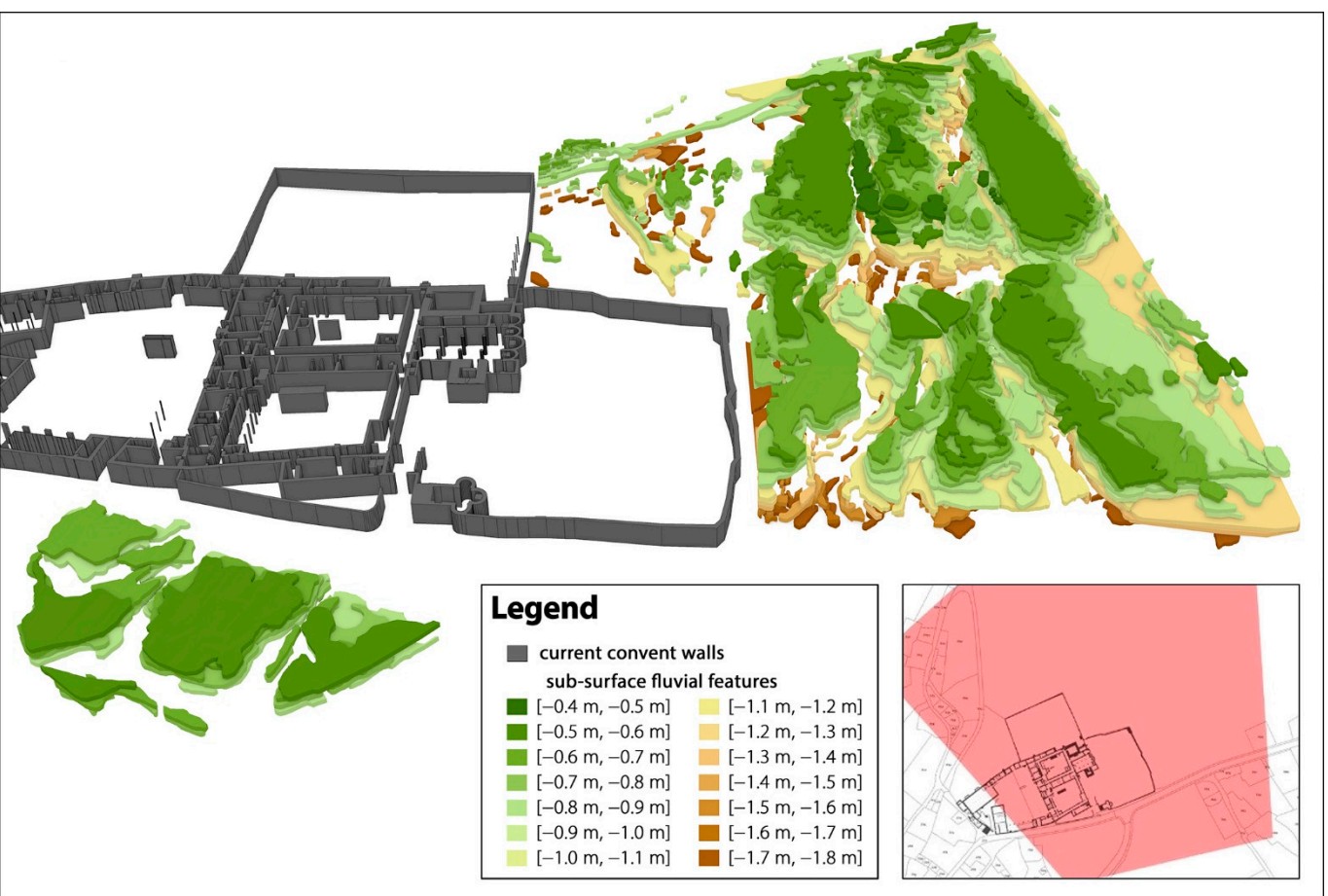

**Figure 6.** The intricate network of deposits related to the Valgarola alluvial fan.

Perhaps the braided stream system developed before the Roman time into a subsurface water flow or smaller, seasonal rivulets that only carried water when it rained or when the snow melted. Even in the 20th century, such open water channels were present to the north and south of the convent [25]. Most likely, these seasonal streams always followed a similar path and were located in the same area where we mapped the more extensive fluvial deposits, because the GPR data suggest that the convent's location deliberately avoids most of these watercourses, as also noted by Goll and Tscholl [25].

*4.3. Drainage, Water Supply, and Irrigation Systems*

The GPR data reveal many narrow and almost parallel, highly reflective linear features (Figure 5B) in the north of survey zone 1. These structures must be drainage channels, as the GPR data depicts the stones that reinforced the banks of such channels and the various sedimentary deposits inside them [25]. These channels intercepted excess water north of the convent and likely directed it to the watercourse on the west side of the building complex. This western watercourse was already active in prehistoric times, as the Bronze Age building excavated in the western courtyard was protected from it by long dry stone walls [26]. More structures were built in Roman times to regulate this dynamic stream [27]. However, its accompanying stone and mudflow deposits continued to exist in the early and high Middle Ages [25]. At present, the stream is a culvert below the Via Döss, a street on the western side of the convent.

The watercourses left and right of the convent reveal that its location was carefully chosen. However, it was still necessary to supply the convent with freshwater. Several artificial channels were identified during archaeological excavations inside and outside of the convent, which indicate the existence of a complex water management system and the presence of a water source north of the convent. One of these channels ran north–south just to the east of the convent's church, feeding water into a large canal in the eastern convent wing. This canal was also supplied from a reservoir in the convent's courtyard. Another channel passed through the length of the western wing of the convent, also coming from the north. In later times, one of these channels supplied water to a pipe that enabled the operation of the watermill in the convent's west court (Figure 5C). The fact that water from the Avigna valley was used to operate the mill is known from an AD 1394 document which mentions an "*aqueductus nostri molendini*", i.e., a raised water channel leading to a mill [28]. The same mill is also mentioned in texts from AD 1462 on the convent's water rights (the latter cited by Goll and Tscholl [25]).

The entirely excavated water mill can be divided into two construction phases. In the first phase (radiocarbon dated to the first half of the 13th century [27]), the mill must have been supplied with water from a stone-lined canal that came from the north. After a fire led to a second construction phase in the 14th century, the renewed mill was operated through a wooden, above-ground water channel [27]. The other end of the mill channel, leading from the mill to the south and probably discharging water or sewage into the Rom river, is a stone-lined structure. It was identified during previous excavations inside the convent and showed up as a reflection in the GPR data in area 3 beneath the road (Figure 7, small parallel features directly to the upper left of A).

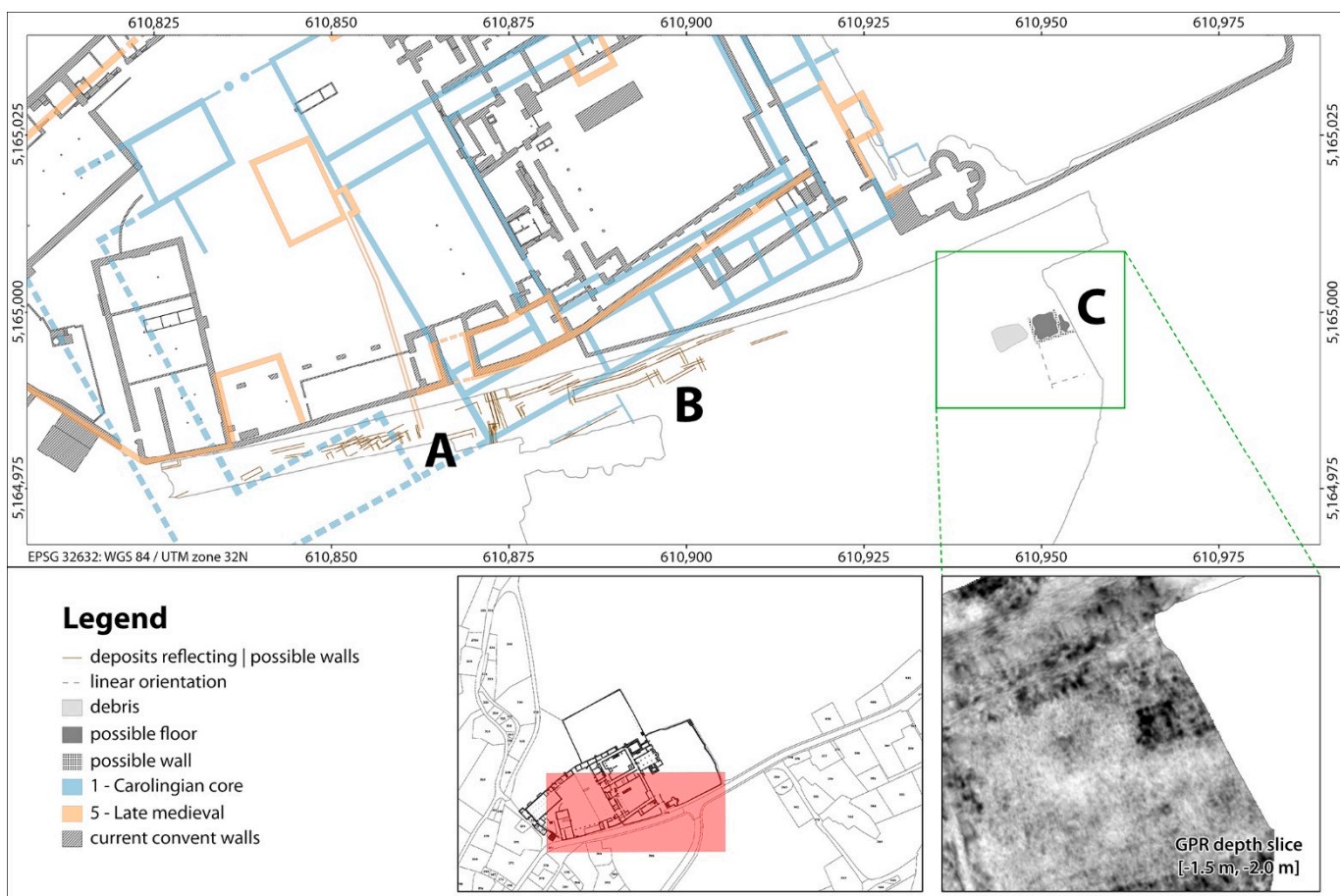

**Figure 7.** Overview of the known historical structures and the prospection additions in area 3 (see the text for **A–C**).

The need for drainage channels in certain parts of this landscape notwithstanding, the dry weather still necessitated the use of complex irrigation systems in other parts [9,29]. Combined with the relatively sheltered location and mild climate of the Müstair valley, these irrigation systems enabled profitable grain cultivation despite the high altitude. The structures in the southeastern part of survey area 1 (Figure 5D,E) are probably related to water-meadow irrigation, a method whereby water from many smaller carriers would overflow onto the meadow or agricultural field. These quasilinear irrigation features are likely (post-) medieval and might have also functioned as field boundaries. Finally, an inoperative irrigation system is also visible on a 1971/1972 map by Furrer and Freund [9] (see the upper right inset of Figure 5). These findings show how the inhabitants reshaped the landscape around the Convent of Saint John and regulated the natural water flows for their specific needs.

### 4.4. Herb Garden

North of the graveyard and east of the convent garden, other linear structures with a width of around 0.8 m appear in area 1 (Figure 5F). They enclose a total area of 0.27 ha divided into two zones, each about 50 m to minimally 21 m. Although hard to interpret, these structures might be connected to the convent because they feature the same primary orientation. They could delineate farming fields owned by the convent but most likely correspond to the extensive herb garden known since at least 1627 [25]. Reflective structures, visible directly below character "F" in Figure 5, border this garden. They are interpreted as rootstock, probably of fruit trees, because fruit cultivation was quite common from AD 100 to AD 1950 in the dry climate of Müstair [30].

### 4.5. Building Structures Connected to the Convent

The construction of the Convent of Saint John can be divided into eight main periods [31] (see Table 1). Building structures identified in the geophysical data can be attributed to most of these phases. The text below presents these structures in approximate chronological order. All illustrations feature a blue-to-red colour map to indicate the known building structures (using the labels of Table 1); this renders them distinct from the newly found archaeological structures, for which less chromatic colours are applied. A striped pattern (or solid dark grey filling) indicates the current convent walls when their building phase is not relevant for the newly found structures depicted in that particular figure.

**Table 1.** Construction periods of the Convent of Saint John (based on [31]).

| Phase | Description | Figure Legend | Time |
|---|---|---|---|
| 1 | Initial Carolingian construction | Carolingian core | around AD 800 |
| 2 | Carolingian building extensions | Carolingian extension | 9th and 10th century |
| 3 | Romanesque Period (bishop Norpert) | Romanesque | 11th century |
| 4 | Bishop Egino | Bishop Egino | 12th century |
| 5 | Late Medieval Period | Late medieval | 13th and 14th century |
| 6 | Abbess Angelina von Planta | Abbess von Planta | 15th and 16th century |
| 7 | Baroque Period | Baroque | 17th and 18th century |
| 8 | Modern time | Modern | 19th and 20th century |

#### 4.5.1. Carolingian and Undated Building Southern Structures

In the south of the convent and beneath the modern street through Müstair (i.e., survey area 3; see Figure 4C), linear features can be detected. Archaeological excavations and exploration trenches dug in the 1980s revealed that this area contains a complex assemblage of features from prehistoric to medieval times [27]. When compared to the archaeological building plan [31], some of these GPR reflections seem to align with the Carolingian and some with medieval and late medieval structures (Figure 7B).

More to the east of survey area 3, a small building of 4.5 m by 2.7 m in an east–west orientation is visible (Figure 7C). At a depth of 1.5 m, a floor and walls can be traced in

the GPR data. The rectangular structure is oriented according to the road and the wall of the graveyard. Purpose and usage are unclear yet, but Sigel mentions a no-longer-existing tower south of the road [2], which likely corresponds with this potential building of which the dating is unclear.

### 4.5.2. Around the Plantatower

The Plantatower is located in the northeast of the convent, named after abbess Angelina von Planta (AD 1478–1509) [31], and dates around AD 957 [32]. This four-floor building was probably built by the bishop of Chur as a fortified residence. The tower is reinforced with a palisade wall and enclosed by a 3.5 m wide ditch. Based on the upper convent garden excavations, this ditch can be traced towards the garden wall [33]. The reflective and absorbing structures in area 1 equal the extension of this ditch and its filling, respectively (Figure 8A). This filling is visible at a depth between 0.65 m and 1.00 m.

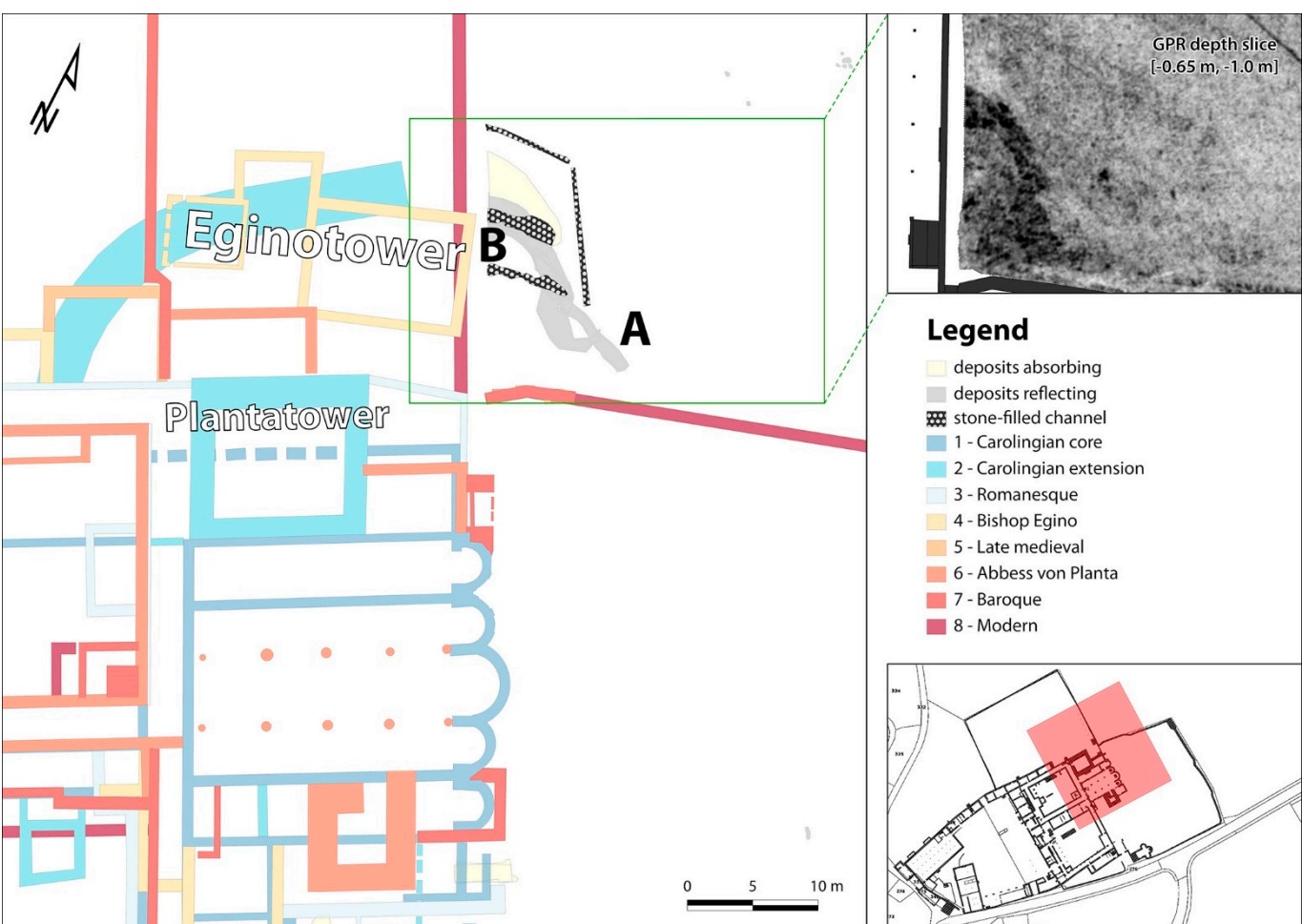

**Figure 8.** Overview of the known historical structures and the new prospection additions around the Planta- and Eginotower (see the text for **A**,**B**).

### 4.5.3. The Eginotower and the Surrounding Building

The eastern wall of the Eginotower, a Bishop's residence from the 12th to 14th century [22], is still partly present under the current garden wall (Figure 8B). This became clear during archaeological excavations in 1993. That same excavation trench also uncovered a larger stone package, which the digging archaeologists hypothetically interpreted as the filling of a channel. The GPR data directly east of the Eginotower supports that thesis and reveals in total four stone-filled channels. They are all positioned above the Plantatower's moat and located at a depth of 0.4 m to 0.7 m below the present surface (Figure 8B). These

stone-filled channels might have served as a substructure for a wooden building adjacent to the Eginotower. Similar structures were excavated in 1993 to the west of the tower and interpreted as a wooden annex [22].

### 4.5.4. The Holy Cross Chapel and Attached Building Structures

The area between the church and the Holy Cross Chapel (Figure 9A) in the southwestern corner of the graveyard contains highly reflective, linear, and rectangular structures (Figure 9-around B and C). Their orientation and location render them as walls or other structures related to the convent, but their dating is problematic. Dendrochronology [32] and radiocarbon dating [34] indicated that the Holy Cross chapel is a Carolingian building constructed around AD 785. Excavations also revealed other constructions from that period around the chapel (Figure 9D). Convent construction activities also continued afterwards [31], with, among other things, the erection of a late medieval ossuary (Figure 9B). The structures in the geophysical data could thus belong to any of these complex construction phases.

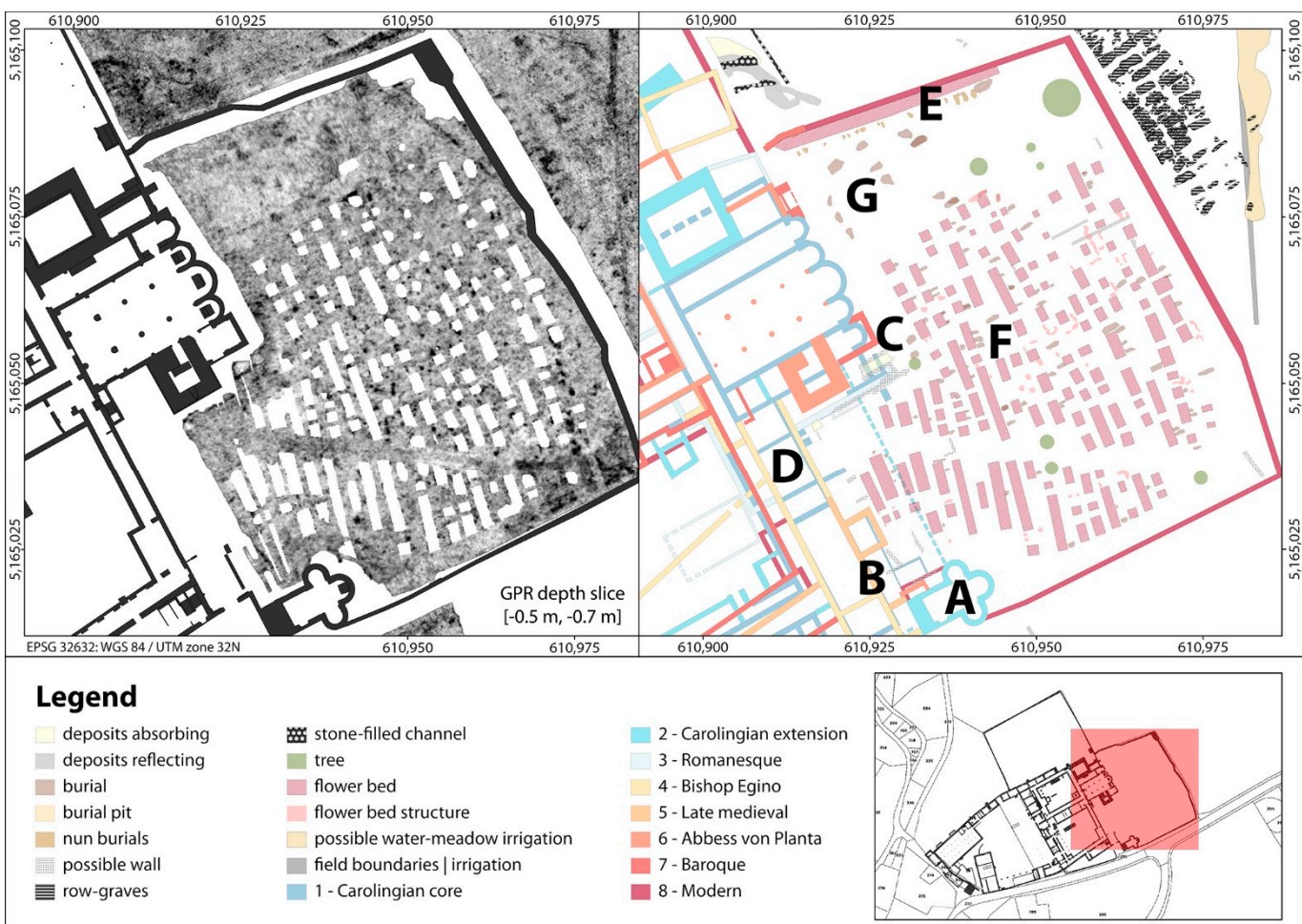

**Figure 9.** Overview of the known historical structures and the new prospection results in and around the present graveyard (see the text to explain the capitalised characters **A–G**).

### 4.6. Burial Grounds

#### 4.6.1. The Present Graveyard

Within the graveyard area (Figure 4B), several excavations have been carried out over the years [35]. These investigations took place mainly in the southwestern part of the graveyard. Although this area currently contains a path connecting the street with

the church, there are still observable structures in the radar data south of the church that correspond to some of those activities (Figure 9C). In total, the archaeological excavations uncovered 665 burials.

Burials in the immediate proximity of churches or chapels were popular in all periods. For this location, it means that skeletons were deposited between the foundation of the convent at the end of the 8th century and the beginning of the 20th century, which equals a timespan of around 50 generations [35]. To the south of the church and the north of the Holy Cross Chapel, a smaller number of Carolingian graves were found. In Romanesque time, no changes in the burial activity in the graveyard area could be detected. In the late medieval times (13th to 15th century), the construction of the southern and western convent wall led to the definition of an actual graveyard and, therefore, defined the new burial area. Eventual burials of the previous periods in that area were destroyed in this process. In the 16th century, most adults were still buried in this graveyard. These burials are all east–west oriented [35].

### 4.6.2. The Nun Graves

Around 1758, a crypt was built as a burial place for the nuns. Four funeral plaques attached to the Chapel of Grace list 93 deceased nuns for the period from 1760 to 1919. In 1964, the crypt was abandoned, and the nuns were reburied along the northern graveyard wall [35]. At least 22 of these burials are still visible above ground, bordering the convent wall on the inside. They are marked with a metal cross and decorated with a flower bed. Over time, these flower beds probably became shortened. They are currently only covering the upper portion of the grave (i.e., the part adjoining the headstone), which allowed the acquisition of geophysical data over the lower part of the grave (Figure 9E).

Consequently, many burial pits are clearly, albeit partly, recognisable in the data. As the same flower bed shortening took place with the other inhumation graves in the centre of the graveyard (see Figure 4B), these burial pits show up in the geophysical data as well (Figure 9F). Graves with orientations different from east–west were not found.

In the open area separating the nun graves from the other graves (Figure 9G), other potential burials can be distinguished between depths of 0.5 m and 0.7 m. These graves are not indicated on the surface. They all have an approximate east–west orientation and an average length of 0.6 m. The northern part of the graveyard along the enclosing wall could not be surveyed, as bushes, trees, and a modern wooden shed occupy this area. Hotz and Mittermair [35] mention that according to priest Sebastian de Capol, victims of the plague from AD 1630 would be buried there.

### 4.6.3. Row Graves

In the northeastern area directly outside the graveyard wall, over forty aligned inhumation burials have been found (Figure 10). Encompassing row graves (*Reihengräber*) in three parallel lines, this burial zone measures circa 48 m by 9 m. The graves show an approximate northeast–southwest orientation but are much longer, broader, and deeper than the graves inside the graveyard. Due to the dense geophysical sampling, it is possible to separate the burial pit from the actual inhumation burial. On average, the grave pits become visible from a depth of 0.5 m and measure 2.6 m by 1.2 m.

The inhumation burials themselves are on average 1.8 m by 0.8 m and appear from an approximate 1 m depth onwards. Scrutiny of the data even allowed potential stone slabs around the burial to be distinguished, which would make these graves very conspicuous. The specific design and arrangement of these row graves are characteristic of early medieval burials, such as those known from the 6th-century Longobards. Originally from Northeast Europe, the Germanic Longobards or Lombards moved from the edge of the eastern Alps to northern Italy in AD 568, bringing along their tradition of inhumation in identically oriented, sometimes stone-lined row graves [36,37].

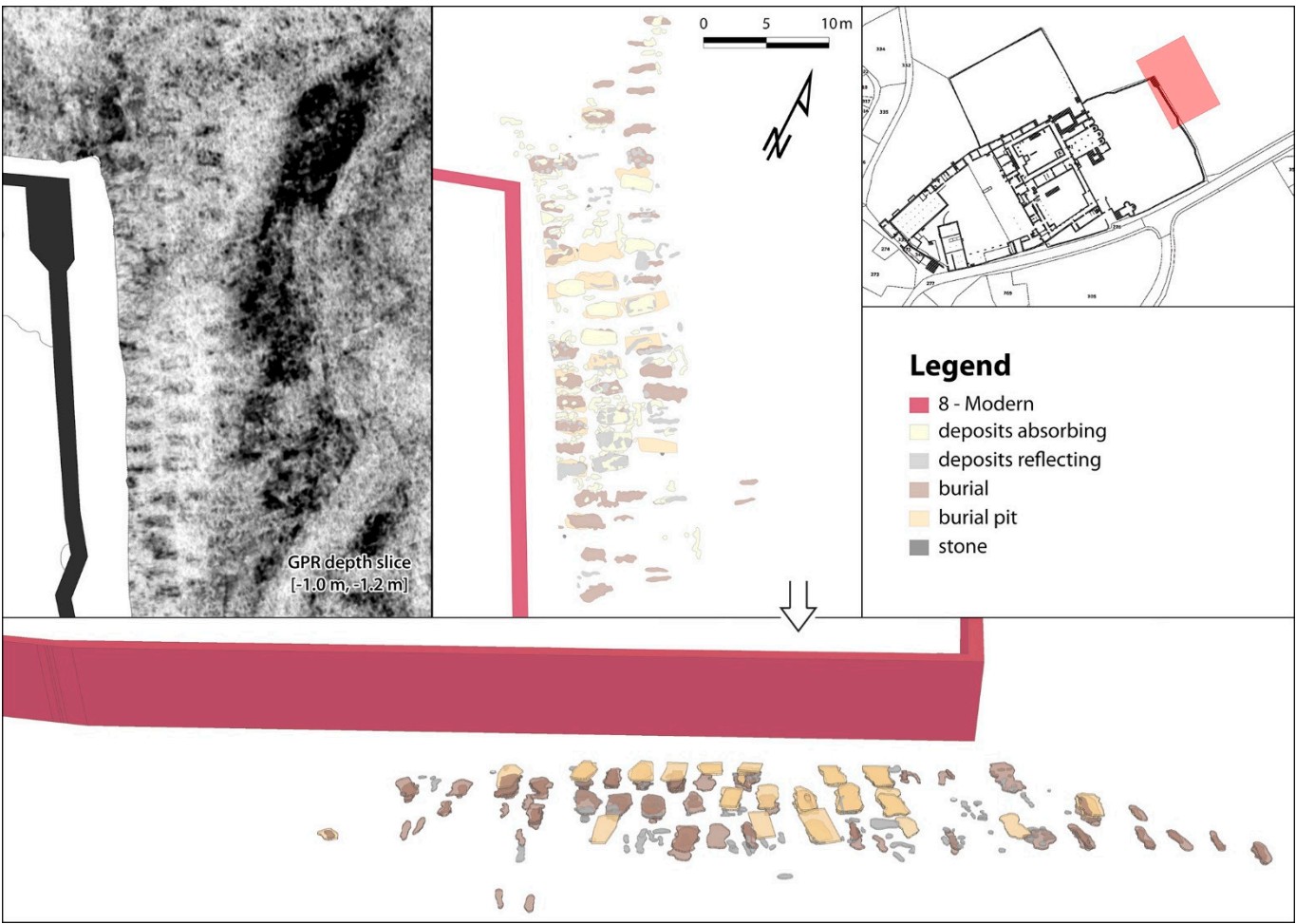

**Figure 10.** Detailed 2D (**upper row**) and 3D view (**lower row**) on the row graves in the northwestern corner outside the modern graveyard. All other mapped archaeological features have been removed for clarity.

Charlemagne conquered the Longobards in AD 774 and became *rex francorum et langobardorum*. This historical event might be seen in relation to the supposed presence of Longobards in a reused Roman hamlet or road station and the subsequent foundation of the convent taking over the control of the important alpine pass leading northwest towards *Curia Raetorum* (present Chur) and Lake Constance [38]. The importance of the southern route across the Umbrail Pass, which would have provided a direct route to the northern Lombard cities, should also not be underestimated [20].

Finally, it is relevant to note that the graves follow the same orientation as the convent's church. If these graves are indeed older than the convent, this might indicate the existence of a predecessor church that determined the orientation of these graves and the later convent church. In the 6th to 8th centuries AD, the construction of small churches in sparsely populated areas and at road junctions can be observed in this region. Such churches usually have burials in and around them [20,39]. This initial church might have been situated underneath the present one, but no excavations have been executed in this area so far. Another possible location for this predecessor church would be underneath the Holy Cross Chapel, where during past excavations, older foundations were identified [40]. Hans Rudolf Sennhauser considered these foundations to be the remains of an older church [41]. In his opinion, this church was a temporary place of worship for the monks who built the convent. Given that these foundations could not be dated, an older date cannot be excluded. Irrespective of the existence and location of such an initial church, the hypothesis of the Longobard graves necessitates additional research in the form of targeted excavations.



## 5. Conclusions and Outlook

This paper reported on a pilot geophysical campaign within and directly outside the walls of the Convent of Saint John at Müstair, Canton of Grisons, Switzerland. The combination of three different geophysical prospection systems with bespoke data processing and mapping tools enabled the documentation of various archaeological and geological structures. Interpreting and dating these structures were often possible thanks to the integration of historical sources, such as maps and excavation reports.

Future research will encompass targeted excavations and more geophysical prospection. First, archaeological excavations planned for 2022 will investigate one or two supposedly Longobard burials outside the recent graveyard and a section of the potential Roman road. Second, additional geophysical surveys will explore the remaining arable fields owned by the convent (Figure 3) and nondestructively investigate the convent garden, the so-called "Kälberwiese" in the northwest of the convent, and the Via Döss in the west. These areas could provide more clues about the prehistoric and Roman past of Müstair.

**Author Contributions:** Conceptualisation: W.N., J.S., and G.J.V.; Formal analysis: J.S., W.N., G.J.V., and P.C.; Funding acquisition: W.N., P.C., and T.R.; Investigation: J.S., G.J.V., A.H., K.L., H.S., and W.N.; Methodology: J.S., W.N., and G.J.V.; Software: A.H.; Supervision: W.N.; Visualisation: J.S. and G.J.V.; Writing—original draft: J.S. and G.J.V.; Writing—review and editing: G.J.V., W.N, P.C., J.S., A.H., T.R., and C.W. All authors have read and agreed to the published version of the manuscript.

**Funding:** This research received no external funding.

**Institutional Review Board Statement:** Not applicable.

**Informed Consent Statement:** Not applicable.

**Data Availability Statement:** Restrictions apply to the availability of these data.

**Acknowledgments:** The surveys were conducted by the Ludwig Boltzmann Institute for Archaeological Prospection and Virtual Archaeology (LBI ArchPro) in collaboration with the foundation "Pro Kloster St. Johann in Müstair" and the Archaeological Service of the Canton of Grisons. The Ludwig Boltzmann Institute for Archaeological Prospection and Virtual Archaeology (archpro.lbg.ac.at) is based on international cooperation between the Ludwig Boltzmann Gesellschaft (A), Amt der Niederösterreichischen Landesregierung (A), University of Vienna (A), TU Wien (A), ZAMG—Central Institute for Meteorology and Geodynamics (A), Airborne Technologies (A), 7reasons (A), LWL—Federal state archaeology of Westphalia-Lippe (D), NIKU—Norwegian Institute for Cultural Heritage (N), Vestfold fylkeskommune—Kulturarv (N), and ArcTron 3D (D).

**Conflicts of Interest:** The authors declare no conflict of interest.

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
