# Peer review of "Prospecting the UNESCO World Heritage Site of Müstair (Switzerland)"

_remotesensing, doi:10.3390/rs13132515_

Round 1
Reviewer 1 Report
In this paper, three geophysical prospection methods were applied for the detection of archaeological features in the UNESCO World Heritage site of Müstair. Although the androgenetic evidences under beneath the surface are evidently detected, the manuscript in this version still have room for a major revision according to the comments as follows:
- The data processing of three geophysical approaches needs to be elaborated in more detail for the technical implementation in other regions with archaeological interest by potential readers. For instance, how to generate a high quality GPR slices as well as 3D tomography? Are there any thresholds recommended?
- What is the impact of soil moisture for the GPR prospection? More quantitively analysis and discussion need to be added in order to highlight the novelty of this manuscript.
- Why the three geophysical methods, including magnetometer, hand-held three-antenna GPR and motorized 16-channel GPR were utilized for this study? What are their merits respectively? In the result interpretation, is there any technical fusion or cross validation applied in order to enhance the robustness of the archaeological discovery. More relevant issues need to be elaborated, I think.
In summary, the manuscript in this version tends to be a scientific report. The novelty of this work, either from methodology or application point of view, need to be highlighted. In addition, quantitative analysis and discussion are lacking.
Reviewer 2 Report
The authors have carefully drafted a paper on an important site which extends our understanding beyond the surface structures and provides useful context for further studies. I only have minor comments concerning the figures. Both the methodology and results are presented in an intelligible and clear manner. It is not a groundbreaking study, but a solide piece of fieldwork which merits to be published in order to be available for future research at the site.
- Please rearrange the figures so their appearance matches the first reference in the text.
- Change the colors on figure 6 - the different shades of brown are barely distinguishable for a reader not intimately familiar with the site
- "Encompassing row graves (Reihengräber)" would perhaps be better translated as "linear cemetery" or described as "rows of graves". I am unaware of the existence of the term "row grave" in English, at least it does not seem to be common. If you want to use it, decide whether you use it with our without hyphen.
